# Salty Twins: Salt-Tolerance of Terrestrial *Cyanocohniella* Strains (Cyanobacteria) and Description of *C. rudolphia* sp. nov. Point towards a Marine Origin of the Genus and Terrestrial Long Distance Dispersal Patterns

**DOI:** 10.3390/microorganisms10050968

**Published:** 2022-05-04

**Authors:** Patrick Jung, Veronika Sommer, Ulf Karsten, Michael Lakatos

**Affiliations:** 1Department of Integrative Biotechnology, University of Applied Sciences Kaiserslautern, 66953 Pirmasens, Germany; michael.lakatos@hs-kl.de; 2Applied Ecology and Phycology, Institute of Biological Sciences, University of Rostock, 18057 Rostock, Germany; veronika.sommer@uni-rostock.de (V.S.); ulf.karsten@uni-rostock.de (U.K.); 3upi Umweltprojekt Ingenieurgesellschaft, 39576 Stendal, Germany

**Keywords:** akinete, Schiermonnikoog, potash tailings piles, biocrusts, pulse amplitude modulation

## Abstract

The ability to adapt to wide ranges of environmental conditions coupled with their long evolution has allowed cyanobacteria to colonize almost every habitat on Earth. Modern taxonomy tries to track not only this diversification process but also to assign individual cyanobacteria to specific niches. It was our aim to work out a potential niche concept for the genus *Cyanocohniella* in terms of salt tolerance. We used a strain based on the description of *C. rudolphia* sp. nov. isolated from a potash tailing pile (Germany) and for comparison *C. crotaloides* that was isolated from sandy beaches (The Netherlands). The taxonomic position of *C. rudolphia* sp. nov. was evaluated by phylogenetic analysis and morphological descriptions of its life cycle. Salt tolerance of *C. rudolphia* sp. nov. and *C. crotaloides* was monitored with cultivation assays in liquid medium and on sand under salt concentrations ranging from 0% to 12% (1500 mM) NaCl. Optimum growth conditions were detected for both strains at 4% (500 mM) NaCl based on morpho-anatomical and physiological criteria such as photosynthetic yield by chlorophyll *a* fluorescence measurements. Taking into consideration that all known strains of this genus colonize salty habitats supports our assumption that the genus might have a marine origin but also expands colonization to salty terrestrial habitats. This aspect is further discussed, including the ecological and biotechnological relevance of the data presented.

## 1. Introduction

Tracing biogeographic patterns of cyanobacteria in the light of modern molecular taxonomic classifications is a traditional focal point in phycology and microbiology. Currently, the available number of cyanobacterial strains, including their DNA sequences that can be categorized according to certain criteria such as specific genera or environmental requirements, e.g., halophilic strains, is by far not completed to a level that allows detailed insights into their spatio-temporal distributions—including their origins.

First attempts for studies on global biogeography were undertaken, for example, for the unicellular hot and cold desert cyanobacterial clade *Chroococcidiopsis* [1]. These authors found no evidence of recent inter-regional gene flow, indicating that *Chroococcidiopsis* populations have not shared common ancestry since before the formation of modern continents. These results indicate that the global distribution of such desert cyanobacteria has not resulted from widespread contemporary dispersal but is an ancient evolutionary legacy [1]. Another example is *Cylindrospermopsis raciborskii*, a toxic cyanobacterium with an invasive bloom-forming nature occurring in water bodies worldwide. This species seems to have its origin in the American continent, but dispersal routes spread first into the African continent and later into Asia and Australia, with Europe being the last continent to be colonized [2]. These are just two examples showing that the scientific community is well aware of the tremendous diversification process that cyanobacteria underwent during their long history that led to morphological, ecophysiological and spatio-temporal differentiation of the phylum.

Besides the well-known cyanobacterial harmful algal blooms (cHABs) that can be monitored from space due to their large distribution [3], cyanobacteria are also common pioneer organisms in desert soils [4], colonizers of Antarctica [5], cave inhabitants [6], symbiotic partners of lichens [7] and are even presumed to be ancient life forms on Mars [8]. This broad spectrum of habitat, geographic distance and abiotic conditions under which cyanobacteria can thrive allows them to define their ecological niche and is reached by their high acclimatization capability. Rapid changes in gene regulation in cyanobacteria that affect the expression of such as the one encoding molecular chaperones, photosynthetic and oxidative stress-related genes followed by complex metabolic adjustments are part of their ecological success story [9,10]. Besides drastic temperature amplitudes [11], long periods of desiccation, strong radiation and pH extremes [12], they are also able to flourish in saline environments [13].

The evolution of effective salt-acclimation mechanisms allowed cyanobacteria to colonize habitats that even shift frequently between low to high salt concentrations, such as brackish environments of estuaries. For model cyanobacteria, salt acclimation strategies are well characterized on the physiological, biochemical and molecular levels. Their basal salt acclimation strategy includes two principal responses, the active export of metabolically harmful ions (mainly sodium and chloride) and the accumulation of compatible solutes via biosynthesis or uptake from the environment [14]. Primarily, the four major compatible solutes sucrose, trehalose (sugars), glucosylglycerol (heterosides, [15]), and glycine betaine (amino acid derivates), are synthesized and/or accumulated by cyanobacteria [14]. These low-molecular mass weight substances are readily soluble in water, can be accumulated in high concentrations and do not interfere with the cell’s metabolism but compensate for changes in the osmotic potential [16]. One example is the strain *Desmonostoc salinum* CCM-UFV059 which showed a salt tolerance when compared to the model *Nostoc* sp. PCC 7120 by accumulating sucrose as the main compatible solute under saline conditions as an acclimatization strategy. *Desmonostoc salinum* CCM-UFV059 was able to accumulate three times the amount of sucrose (up to 100 nmol mg dry weight^−1^) after 24 h of exposure to 250 mM NaCl [17]. Similar values have been reported for cells of *Nostoc muscorum* PCC 7119 after 8 days of exposure to 200 mM NaCl [18], while *Synechocystis* sp. PCC 6803 exposed to 200 mM NaCl for 24 h contained only one-fifth of sucrose [19]. For unicellular freshwater strains, it was, for example, shown that most of them were able to grow at 250 mM NaCl but unable even to survive 500 mM NaCl, while phylogenetically related isolates from saltwater environments started to lose vitality only around 1000 mM NaCl [20]. However, as main osmolytes, sucrose or trehalose is found in many freshwater strains with a limit of 600 mM NaCl, while glycine betaine or glutamate betaine is found in halophilic cyanobacteria with a limit close to 3000 mM NaCl [14]. Besides the mentioned four main compatible solutes produced by cyanobacteria, there can also be some more unusual substances involved in osmotic acclimation, such as the recently identified homoserine betaine in *Trichodesmium erythraeum* [21].

Despite marine environments, such salt-tolerant cyanobacteria can also be found in salt-rich terrestrial or even artificial situations. Recently high throughput sequencing of biocrusts on potash tailings piles in Germany uncovered a vivid microbial community under high salt conditions [22], with the filamentous orders Synechococcales and Oscillatoriales as the most dominant cyanobacteria [23]. Several cyanobacteria belonging to the filamentous clade *Phormidesmis* could be isolated in addition to a strain of the heterocytous genus *Cyanocohniella* [24]. This genus comprises the type species *C. calida* that was isolated from a hot water spring (Czech Republic; [25]) and in addition *C. crotaloides* from a sandy beach of a North Sea barrier island (The Netherlands; [26]) as well as *C. hyphalmyra* from a brackish lake (Greece; [27]). All of them have in common that they are unusually unstable in their morphological traits, with a life cycle that is among the most complex observed in all cyanobacteria [25]. The similarity of the various stages to existing genera in diverse subclasses is especially problematic, with a *Pseudanabaena*-like stage, *Nostoc*-like stage, and *Chlorogloeopsis*-like stage accompanied by the formation of hormogonia, akinetes and heterocytes that can have a cryptic character.

Despite the fact that this morphological variety can be problematic in terms of identification, it simultaneously offers a great opportunity to detect morphological changes under varying abiotic conditions. For these reasons, we herein described that the strain *Cyanocohniella* sp. SY-1-2-Y was previously isolated from the potash tailings piles surroundings [24] as a new species and tested its salt tolerance in comparison to *C. crotaloides* isolated from the coast of the North Sea. Morphological and molecular work in the context of the so-called polyphasic approach on this strain helped to confirm the description of a novel species based on its 16S rRNA phylogeny according to the requirements of the International Code of Nomenclature for algae, fungi, and plants [28]. Since the vast majority of studies on cyanobacterial salt acclimation have been carried out on unicellular strains such as *Synechocystis* sp. PCC 6803 [29]—which cannot fix atmospheric N_2_—we here aim to close this gap by comparing the salt tolerance of the heterocytous *Cyanocohniella* sp. SY-1-2-Y with the related species *C. crotaloides*. In addition, this also gave hints for a possible origin of the genus and its adaptation and acclimatization potential with respect to its biogeography.

## 2. Material and Methods

### 2.1. Origin of Strain and Isolation

The strain SY-1-2-Y originates from a former potash mining site in Schreyahn (52°55′89″ N, 11°4′62″ E), Lower Saxony, Germany. Three isolated potash tailings pile residues remained from the waste of the fertilizer production that was conducted till the 1930s. The cavities of the former pit ‘Rudolph’ were flooded by groundwater and thus created a pond. The potash tailings piles with a height of up to 1.5 m [28] are situated between farmland and a small forested area. Due to the occurrence of many facultative as well as obligate halophytes, this site is now part of a protected area. A recent study described well-developed biocrusts in the close surrounding as well as young green-algal biocrusts on top of the tailings or at the pile bottoms [24]. The strain SY-1-2-Y was isolated from a well-developed biocrust dominated by green algae with the presence of mosses, as described before [24].

For comparison of morphological features and salt tolerance, the cyanobacterial strain *Cyanocohniella crotaloides* PJ S45 was used that was isolated in May 2018 from microbial mats located at the lowest part of the supratidal to the upper part of the intertidal zone of a sandy beach of the North Sea barrier island Schiermonnikoog (53°28′23.62″ N, 6°8′32.47″ E) as described by [26]. The beach has a typical hydrological zonation with an upper saline plume, terrestrial groundwater discharge from the island aquifer, and a saltwater wedge. The mats were influenced by sea water during high tide as well as by rain, storms, floods, upwelling freshwater, and ice covers. These physical forces sporadically led to the destruction of the microbial mats that form throughout the year.

### 2.2. Molecular Characterization

The cells of two-week-old cyanobacterial culture were disrupted by shaking with 1.25–1.55 mm glass beads in combination with a threefold freezing and thawing cycle (liquid N_2_, heating block 65 °C). The DNA was extracted with the NucleoSpin Plant II mini kit (Macherey Nagel, Düren, Germany) following the manufacturer’s instructions. PCR was performed as previously described using the primers SSU-4-forw and ptLSU C-D-rev for cyanobacteria. Sanger sequencing was conducted by Genewiz (Göttingen, Germany) using the primers SSU-4-forw, Wil 6, Wil 12, Wil 14, Wil 5, Wil 9, Wil 16 and ptLSU C-D-rev [29,30].

The generated sequences were assembled with Geneious Prime (v2021.0.1; Biomatters Limited, Auckland, New Zealand) and submitted to the National Center for Biotechnology Information (NCBI) GenBank, as stated in the species description, with accession number MT946558. 

The assembled 16S rRNA gene sequence obtained from the isolate and related sequences of cyanobacterial strains cited from GenBank were used for phylogenetic analyses, including *Gloeobacter violacaeus* as the outgroup for the 16S rRNA gene alignment, applying the Muscle algorithm in Mega X [31]. Finally, 108 nucleotide sequences from nostocalean strains were used for the phylogenetic comparison, including 1501 bp of the 16S rRNA gene. The evolutionary model that was best suited to the database used was selected on the basis of the lowest akaike information criterion value and calculated in Mega X using the evolutionary model RGT + G + I of nucleotide substitutions. The maximum likelihood method (ML) with 1000 bootstrap replications was calculated with Mega X and Bayesian interference (BI) phylogenetic analyses, with two runs of eight Markov chains executed for one million generations with default parameters with MrBayes 3.2.1 [32]. Each analysis reached stationarity (average standard deviation of split frequencies between runs < 0.01) before the end of the run.

Models of the secondary structure of the 16S–23S ITS gene region of the isolate were built in comparison to phylogenetic or morphologically related species. Helices were folded with default settings using the online software RNAstructure Web Server [33].

### 2.3. Morphological Characterization

The strain’s life cycles were studied on BG 11 agar plates as well as in liquid BG 11 medium by light microscopy to ensure collecting a comparable maximum of the strain’s morphological features. Morphological characteristics were evaluated with a Panthera KU trinocular light microscope (Motic, Barcelona, Spain) coupled with a MicroLive Multi Format camera and the software MicroLive v4.0 (C. Linkenheld, Oppenau, Germany). About 200 cells were measured, and a mean size range is given, reflecting the variability of irregular cell sizes. In addition to bright-field images, samples were analyzed under autofluorescence and ACN staining. The latter is a 20:1:1 mix of Astra Blue, chrysoidin and Neufuchsin (0.1 g Astra Blue in 79.5 mL H_2_O and 2.5 mL acetic acid, 0.1 g chrysoidin in 100 mL H_2_O and 0.1 g Neufuchsin in 100 mL H_2_O) that allows differentiation of structures according to color due to the binding characteristics of the substances. Acid mucopolysaccharides are stained blue by Astra Blue, while cellulose or lignin is stained red by Neufuchsin and hydrophobic substances such as cutin yellow. 

In addition, the strain was checked for its ability to fix nitrogen from the air by cultivating it separately on BG 11 agar plates without nitrogen for about 2 months. All agar plates were sealed with parafilm to prevent the desiccation of the agar and the cells.

For a detailed comparison of morphological features, all the above-described analyses were also carried out for *Cyanocohniella crotaloides* PJ S45 as a closely related species of the same genus.

### 2.4. Holotype Preparation

The new species was described following the rules and requirements of the International Code of Nomenclature for algae, fungi and plants [28]. For the preservation of the type strain, young, 3-week-old cultures were transferred into 5-mL glass bottles with a 4% (*v*/*v*) formaldehyde–water mixture. The preserved material is deposited at Herbarium Hamburgense, Hamburg, Germany, with the deposition number HBG-025346, while the living culture material is given to the German Collection for Microorganisms and Cell Cultures (DSMZ), Braunschweig, Germany, with the deposition number DSM 113757.

### 2.5. Salt Assays

The strain SY-1-2-Y and *Cyanocohniella crotaloides* PJ S45 were grown in 200 mL liquid BG 11 medium till they reached the stationary phase in a culture cabinet (CLF Plantclimatics, Percival; Wertingen, Germany) at 20 °C with a light:dark cycle of 18:6 h and a photon fluence rate of 30 μmol m^−2^ s^−1^. Both strains tend to grow in cellular aggregates making a uniform distribution of cells for experiments challenging. In order to break up these cell aggregates, the liquid cultures were placed in an ultrasonic bath until the aggregates were separated. Cell wall integrity was checked afterward by microscopy to ensure minimal membrane damage during sonication. Cells were then centrifuged at 1000× *g*, the supernatant was discarded, and wet biomass was divided into portions of 150 mg that were collected in 1.5 mL tubes each. The biomass was resuspended in 500 µl of BG 11 of varying salt concentrations (0.5% = 62.5 mM, 0.75% = 93.75 mM, 1% = 125 mM, 2% = 250 mM, 4% = 500 mM, 12% = 1500 mM) and BG 11 without salt as a control. Of these suspensions, 150 µl were transferred into 12 well plates filled with 3 mL of each of the BG 11 salt media in varying concentrations in triplicates, hereafter referred to as the ‘liquid salt assay’. In addition, 160 µl of the suspensions were transferred to 12 well plates filled with 3 g of sterile quartz sand, hereafter referred to as the ‘sand salt assay’. To each well-containing sand and algal suspension on the surface, 3 mL of each BG 11 salt solution was given. This sand salt assay was carried out to check for differences in the growth with respect to the terrestrial origin of both the isolate SY-1-2-Y and *Cyanocohniella crotaloides* PJ S45. Finally, a positive control containing liquid BG 11 medium without salt but with biomass was added to the liquid and the sand assay as well as a negative control containing only BG 11 medium without salt and biomass in order to check for contaminations. The assays were sealed with parafilm and cultivated for eight weeks under the conditions described above. 

The liquid salt assay was evaluated by autofluorescence and light microscopy in order to check for morphological changes in the strains depending on the salt concentrations, such as akinete formation and filament length.

### 2.6. Physiology

The physiological characteristics of the sand salt assay were determined after 8 weeks in the 12 well plates via non-invasive chlorophyll-*a*-fluorescence using pulse amplitude modulated equipment (Mini-Head, IMAGING-PAM, H. Walz, Effeltrich, Germany) to capture the ecophysiological viability state of the cells. Parameters of maximum quantum yield of PSII (*F_v_/F_m_*) and quantum yield of PSII (*Φ_PSII_* = (*F′_m_ − F_t_*)/*F′_m_*) were derived according to Genty et al. [34]. Ambient light was excluded during the measurements, and the samples were dark adapted by a piece of black cloth for 30 min before measuring. The measured photo-physiological parameters were tested for normality (Shapiro–Wilk test) and compared between salt concentrations by ANOVA using the Tukeys HSD post hoc test (Table 1).

## 3. Results

### 3.1. Phylogeny Based on the 16S rRNA Gene

The phylogeny that is based on the 16S rRNA gene region depicted in Figure 1 shows that the strain SY-1-2-Y clusters within the genus *Cyanocohniella* framed by the genera *Halotia* and *Nodularia*. The 16S rRNA gene sequence of the isolate is highly related to those of other *Cyanocohniella* species and isolates and shares 99.18% similarity with the strain SY-1-2-EE from the same habitat and 98.9% with strain ACSSI 322 from Russia as the closest related representatives. Together these three form a separated and well-supported cluster while the type strain of the genus *C. calida* Kastovsky 1996/2 (98.56%) as well as *C. crotaloides* PJ S45 (98.05%) and *C. hyphalmyra* TAU-MAC 3117 (98.44%) form an additional cluster.

The predicted Box B and D1-D1′ secondary structures of the 16S-23S rRNA ITS region are highly comparable to those of the above-mentioned described species (Figure 2). However, the top base of the D1-D1′ structure is different from all other strains.

### 3.2. Strain Morphology and Nitrogen Fixation

Microscopic analyses showed that the new strain has a highly variable morphology caused by differentiation during various life stages that comprise a uniseriate *Nodularia*-like morphology with straight to slightly curved trichomes, a *Nostoc*-like stage with smaller, curved trichomes that are embedded in a common sheath and a motile *Pseudanabaena*-like stage with straight trichomes and apical cells (Figure 3). In addition, the formation of ellipsoidal akinetes was frequently observed (Figure 3).

Strain SY-1-2-Y showed vivid growth on nitrogen-free BG 11 medium and frequently formed rounded to slightly compressed heterocytes, while *Cyanocohniella crotaloides* PJ S45 showed little growth on nitrogen-free BG 11 medium, which was also indicated by the pale olive green to yellow coloration of the biomass on the agar (Figure 4). Here, heterocyte formation could be observed as well, but the heterocytes appeared to be hampered in function visible by collapsed cell contents of the cells (Figure 4E,F arrows).

### 3.3. Taxonomy

A taxonomic evaluation of the isolated strain in the context of the polyphasic approach [35] and additional ecologically based analyses allowed the establishment of the new species *Cyanocohniella rudolphia* sp. nov. SY-1-2-Y. In the following section, the resulting taxonomic treatment will be depicted as a consequence of the above-explained findings regarding its phylogenetic position, morphology and ecology.

#### 3.3.1. *Cyanocohniella rudolphia* sp. nov. P. Jung et V. Sommer

##### Description

Populations of *Cyanocohniella rudolphia* sp. nov. show a polymorphic life cycle. This species forms bright green to blue-green heterogeneous mats on agar associated with a liquid and slimy mucilage. The adult and main form resemble *Nodularia*-like structures with straight, constricted and parallel always arranged uniseriate filaments embedded in a non-lamellated, fine and colorless sheath. During this stage, cells are rounded to ellipsoidal, flattened at the cross walls, 3–4.5 µm long, 2.8–3.5 µm wide. In addition, the uniseriate filaments can also become densely curved, resembling a Nostoc-like morphology embedded in a well-pronounced, non-lamellate, hyaline sheath of up to 13 µm thickness. At this stage, cells are smaller than those of the *Nodularia*-like stage, isodiametric with a diameter of 2.1–3.5 µm. Heterocytes also are present on nitrogen-containing medium, which are ellipsoidal and wider (3.8–5.3 µm) than long (3–4 µm) and intercalary. Akinetes are formed during the *Nodularia*-like stage in uniseriate series as oval, mostly granulated blue-green to brown cells with a thick cell wall that can be up to 7.5 µm long and 4.1 µm wide and can sometimes be surrounded by a massive sheath layer. After serial akinete formation, the uniseriate akinete containing filaments detach. From the akinetes young filaments or motile hormogonia develop that show a *Pseudanabaena*/*Leptolyngbya*-like morphology with constricted cells that are cylindrical and 2.4–3.8 µm long and 2.7 µm wide. The apical cells of these filaments are longer (up to 5.3 µm) than wide (2.7 µm) and conical. During this stage, cells are granulated. Cell division takes place in one plane during all stages. Thylakoid position is coiled as the cell content is homogeneously distributed across the mostly non-granulated blue-green cells.

##### Habitat

Biocrusts with moss in potash tailings pile surroundings at Schreyahn, Saxony-Anhalt, Germany (52°55′89″ N, 11°4′62″ E).

##### Etymology

‘*Rudolphia*’ is the name of the respective potash pit ‘Schacht Rudolph’, belonging to the former potash works ‘Teutonia’, Schreyahn.

##### Type Locality

Schreyahn, Lower Saxony, Germany (52°55′89″ N, 11°4′62″ E).

##### Holotype

The preserved holotype specimen of the cyanobacterium is available via the Herbarium Hamburgense, Hamburg, Germany (HBG-025346). This was prepared from the living strain which was the source of the 16S-23S ITS rRNA gene sequence deposited at GenBank with accession number MT946558.

##### Reference Strain

*Cyanocohniella rudolphia* sp. nov. SY-1-2-Y (DSM 113757).

##### Phylogenetic Relation and Secondary Structure of the 16S-23S ITS Gene

*Cyanocohniella rudolphia* sp. nov. shares 98.9% identity with the 16S rRNA gene sequence of strain ACSSI 322 from Russia as the closest related representatives. Together they form a separated and well-supported cluster while the type strain of the genus *C. calida* Kastovsky 1996/2 (98.56%) as well as *C. crotaloides* PJ S45 (98.05%) and *C. hyphalmyra* TAU-MAC 3117 (98.44%) form an additional cluster. The secondary structures differ only by the top base of the D1-D1′ domain.

##### Differentiation against Other Species

*Cyanocohniella rudolphia* sp. nov. can be differentiated by uniseriate serial akinete formation while *C. crotaloides* forms multi-serial akinetes; akinete formation was not observed for *C. hyphalmyra*. In addition, the adult form of *C. rudolphia* resembles those of *Nodularia*-like morphology with straight uniseriate filaments, while the adult development of *C. crotaloides*, *C. hyphalmyra* and the type strain for the genus *C. calida* resembles a *Chlorogloeopsis*-like morphology.

### 3.4. Salt Assays

The salt tolerance of both species, *Cyanocohniella crotaloides* PJ S45 and *C. rudolphia* sp. nov. SY-1-2-Y was tested in a liquid salt assay with BG11 and in a sand salt assay where they were cultivated on the surface of sand with the addition of liquid BG11 medium under varying salt conditions (0%, 0.5%, 0.75%, 1%, 2%, 4% and 12% NaCl). 

Both species grew at BG 11 without any NaCl (Figure 5), but akinete formation and some cells depleted in pigments could be observed by light microscopy and fluorescence microscopy (Figure 6A–C,G–I). Weak growth patterns and olive-green to brown biomass coloration were especially observed for *C. rudolphia* sp. nov. SY-1-2-Y at the sand salt assay (Figure 5).

Best growth patterns for both species were observed at 4% NaCl (500 mM) in both the liquid and the sand salt assay, where biomass growth was highest, most uniform and vividly green to blue-green coloration indicating optimum conditions (Figure 4). At 4% NaCl, filaments of both species were long, intensely colored and with a low degree of akinetes or pigment-depleted cells as observed by microscopy (Figure 6D–F,J–L). Moreover, the mean photosynthetic quantum yield of PSII (*ΦPSII*), measured by chlorophyll *a* fluorescence (Table 1), increased from around 0.2 to 0.3, with ascending salt concentration from the control to 2% NaCl reaching the highest and most significant photosynthetic performance at 4% (Tukeys HSD, *p* < 0.05). The photosynthetic performance did not significantly differ between both strains.

At 12% (1500 mM) NaCl, *C. rudolphia* sp. nov. SY-1-2-Y died off in the liquid salt test, but biomass weakly remained at the sand assay without any signs of growth. In contrast, *C. crotaloides* PJ S45 did not increase biomass but survived during the time of the experiment in both assays (Figure 5). Both strains revealed poor photosynthetic quantum yields (*ΦPSII*) lower than 0.05 and were significantly different at 12% NaCl (Tukeys HSD; *p* < 0.01; Table 1) in comparison to all other salt concentrations of the sand assay (Table 1). The latter mainly formed olive green to brownish filaments in liquid BG 11 medium at 12% NaCl and, to a great extent, akinetes were observed by light microscopy (Figure 6O). This was also supported by fluorescence microscopy, where a high amount of cells without or with degraded pigments could be detected (Figure 6M,N).

## 4. Discussion

### 4.1. Implications for Taxonomy

Recently, a few cyanobacterial strains were isolated from highly saline German potash tailing piles that spanned from Synechococcales and Oscillatoriales over to nostocalean taxa [36]. The newly described heterocytous species *Cyanocohniella rudolphia* was isolated from a biological soil crust-like top layer and can be differentiated from other closely related taxa based on its 16S rRNA gene phylogeny, secondary structures of the ITS and morphological characteristics. Interestingly, *C. rudolphia* sp. nov. SY-1-2-Y formed a separate cluster with the undescribed strain ACSSI 322 from an arable alkaline, saline meadow in Russia in the distance to the previously described type strain *C. calida* as well as *C. crotaloides* and *C. hyphalmyra* (Figure 1). Based on a morphological point of view, a taxonomic split into two different genera based on this cluster formation is not indicated yet, since all of the members of this genus share similar characteristics. In addition, all described species have high similarity in their 16S rRNA gene region (>98%) and their secondary structures (Figure 2), which also does not justify a split.

The most prominent morphological feature of the genus lies in the high variability of cell forms, types and sizes as a consequence of one of the most complex life cycles known for cyanobacteria so far (Figure 3). Besides vegetative cells placed in uni- or multi-seriate, straight to coiled and twisted filaments, akinetes and heterocytes can also be developed. Together they can be arranged resembling *Nostoc*, *Chlorogloeopsis* or *Pseudanabaena*. In addition, *C. rudolphia* sp. nov. also forms *Nodularia*-like straight uniseriate filaments as a unique discrimination feature. In addition, *C. calida*, *C. crotaloides* and *C. hyphalmyra* have been isolated from aquatic environments, while *C. rudolphia* sp. nov. is the first from a terrestrial habitat. Initially, the genus *Cyanocohniella* was presumed to be a thermotolerant genus because the type strain *C. calida* was isolated from a 55 °C hot mineral spring [25]. However, this has been excluded by Jung et al. [26] for the description of *C. crotaloides* from intertidal beach mats as well as by Panou and Gkelis [27] for the isolation of *C. hyphalmyra* from brackish water. However, the thermotolerance of the genus has never been tested. 

### 4.2. Salt Tolerance

Most green algae, diatoms, dinoflagellates and other eukaryotic phototrophic microorganisms generally perform very poorly at salt concentrations above 10%, while a broad spectrum of cyanobacteria often forms dense, vivid aggregates with high photosynthetic activities [22,37,38]. However, a great proportion of halophilic cyanobacteria is restricted to the aquatic environment such as the oceans, shallow hypersaline lakes or saltern evaporation ponds. Prominent unicellular examples are the *Aphanothece halophytica* complex, which is also known by other taxonomic names such as *Halothece*, *Euhalothece*, *Coccochloris elabens* or *Cyanothece*, and share optimum growth conditions at a salt concentration of around 6–15% [39]. Furthermore, several strains of the non-heterocytous, filamentous species *Coleofasciculus chthonoplastes* (formerly known as *Microcoleus chthonoplastes*) were isolated from salt-rich aquatic environments that even exceeded 20% NaCl concentrations [40]. However, findings of terrestrial cyanobacteria under hypersaline conditions are rare and mostly relate to halite evaporite rocks from the driest part of the Atacama Desert, Chile [41,42]. In contrast, Cumbers and Rothschild [20] investigated the salt tolerance of unicellular *Chroococcidiopsis*-like strains and among them were also some terrestrial strains. Most of the strains that were tested could not grow at 4% (500 Mm) NaCl concentrations, including the terrestrial strains CCMEE 029 (soil, Negev Desert, Israel), UTEX-CCMEE 246 (soil, Negev Desert, Israel), *C. cubana* SAG 39.79 (soil, Cuba) and CCMEE 171 (Antarctica). While 4% NaCl marked the salt limit for those strains, it turned out in our study that this was the optimum salt concentration for the growth of *Cyanocohniella crotaloides* and *C. rudolphia* in liquid BG 11 medium and on sand. At 4%, NaCl both strains were most vivid, as indicated by quantum yield measurements of PS II, where the highest photosynthetic yield could be found (Figure 5). Moreover, fluorescence microscopy and light microscopy revealed that the populations in the liquid salt assay were most intact with little akinete formation (Figure 6) as a sign of unfavorable culture conditions. During comparable studies, *Fremyella diplosiphon* and also *Anabaena cylindrica*, for example, showed morphological abnormalities induced by salt that were most likely caused by salt-induced osmotic stress [43,44]. For *Synechococcus* sp. PCC 7942, for example, it was reported that salt concentrations of 500 mM NaCl triggered cell elongation and resulted in the bending of cells [45]. These results are in agreement with the observations of [46,47] in *E. coli* and *Lactobacillus casei*, respectively, where salt stress also triggers cell size elongation.

Most other *Chroococcidiopsis*-like strains from marine environments in the study of Cumbers and Rothschild [20] still showed growth at 8% (1000 Mm) NaCl, but only CCMP 1991 (Hawaii, USA) exhibited some cells that at least survived under 12% (1500 Mm) NaCl. Therefore, it is remarkable that *C. crotaloides* was alive under 12% NaCl during the liquid and the sand salt assays, and *C. rudolphia* showed at least physiological activity on sand.

Low salt tolerance has been reported for some terrestrial strains of *Nostoc commune* [48] but has often been linked to extracellular polymeric substances (EPS) that are involved in acclimation processes toward increasing salt concentrations due to osmolytes that can be excreted into the EPS and re-absorbed when needed [49,50]. However, in general, EPS production significantly varies in terms of quality and quantity in liquid culture compared to cultivation on sand [51,52]. Thus, salt assays should be performed not solely in liquid media—especially not when terrestrial strains are the objectives of the study. Although the results of the liquid and sand salt assay were almost identical, we found a significant difference for *C. rudolphia* where no cells remained under 12% NaCl in the liquid salt assay, but physiologically active cells survived the same salt concentration on sand (Figure 6). This could be linked to metabolic changes as a consequence of the cultivation on sand that leads to a higher EPS amount and composition, thereby protecting the cells from osmotic shock at least to a higher degree than under liquid conditions.

### 4.3. Ecological Importance and Biotechnology

In order to increase sustainability, the use of large amounts of limited freshwater resources for large-scale cultivation of cyanobacteria for biotechnological production of biofuels or bulk chemicals should be avoided. Instead, mass cultivation of cyanobacteria should preferably be carried out in brackish or sea water [53]. The influence of salinity on cyanobacterial productivity and specific substances of high value is scarcely investigated, but strains that can potentially grow in sea water offer new opportunities for biotechnology. Here, some compatible solutes, such as the heteroside glucosylglycerol can be used as a stabilizing agent for enzymes or antibodies and hence enables their long-term storage in freezers [54,55] or is used in skin revitalizing creams (e.g., bitop AG, Germany, and designated as Glycoin^®^). With respect to productivity, both *Cyanocohniella* strains investigated are promising taxa as they reached the best vitality under 4% NaCl, while the model strains widely used in biotechnology *Synechocystis* sp. PCC 6803 and *Synechococcus* sp. PCC 7002 show decreased production rates of glucosylglycerol at such salinity [16].

Besides biotechnology, such salt-tolerant terrestrial strains can play significant roles in the remediation of (salt-affected) soils where they can restore the function of the overall microbiome and prevent erosion or desertification [56]. For these purposes, often the non-nitrogen fixing, filamentous cyanobacterium *Microcoleus* spp. is used [57], which shows great abilities in concatenating soil particles as a pioneer microbe. Recently, similar approaches have been undertaken with eukaryotic green algae that were used as inoculum in order to colonize and stabilize potash piles [58]. However, the two *Cyanocohniella* strains are also suitable for such purposes and additionally have the advantage that they can fix atmospheric nitrogen, which enriches soil fertility and thus can speed up remediation processes.

### 4.4. Implications for Biogeography

Given the high number of species richness and diversity studies with a focus on cyanobacteria across the world, it remains remarkable that a genus with such a conspicuous morphology as *Cyanocohniella* has been found only a few times until now—and always linked to salt-rich conditions: The type strain *C. calida* is isolated from a hot mineral spring, *C. crotaloides* from a marine sandy beach, *C. hyphalmyra* from brackish water, strain ACSSI 322 from saline soil, an undescribed *Cyanocohniella* species from the walls of a mineral cave (Jung et al., in preparation) and *C. rudolphia* from potash tailings heaps (Figure 7). Salt seems to define the ecological niche of this genus; thus, the genus itself might be a good starting point to reconstruct the lineages’ origin and trace its spatio-temporal distribution pattern in the future. One scenario could be that the genus has its origin in the marine environment, and the ecophysiological acclimatization potential of cyanobacteria, in general, allowed them to establish under a broad spectrum of habitats with the restriction of salt being available to a significant proportion. As a consequence, the halophilic and ‘aquatic, marine’ genus was presumably able to conquer salt-rich terrestrial (micro)ecosystems such as biocrusts on potash tailings heaps, cave walls and saline arable soils.

This theory for the genus *Cyanocohniella* is speculative but can also be supported by the finding of a relatively short distance from the collection points of the strains and the distance to the marine environment (Figure 7). *Cyanocohniella calida* has the greatest distance to the sea spanning about 900 km, but this is well in the range of, e.g., migratory birds or wind dispersal [59] that might have transported cells from the sea to the terrestrial site. The chances for long-distance dispersal of the genus are also increased by the ability to produce akinetes as resting stages.

In principle, the distribution of microbes across the world should be based on each organism’s potential to be transported to a new habitat by any method at any moment (past, present or future) rather than on the current condition. Once an organism has established itself as a cryptic species in a new environment, its speciation will be influenced by two factors: (1) the distance between the new environment and the original source, which affects the rate of supply of new ‘original-like’ organisms, and (2) the local microbial (cyanobacterial) dynamics, which affect the amount of time an organism has to evolve. According to the ubiquity hypothesis [60], the vast population sizes of microorganisms drive ubiquitous dispersal (‘everything is everywhere, but the environment selects’, [61]) and make local extinction virtually impossible [62].

## 5. Conclusions

Here we presented a novel theory about the spatio-temporal distribution patterns and the origin of the nostocalean cyanobacterial genus *Cyanocohniella* and its linkage to salt tolerance. This assumption might be resolved if the genomes of members of the genus are made accessible or when new strains are found in the future; thus, we want to encourage the scientific community to follow up on these initial hypotheses.

## Figures and Tables

**Figure 1 microorganisms-10-00968-f001:**
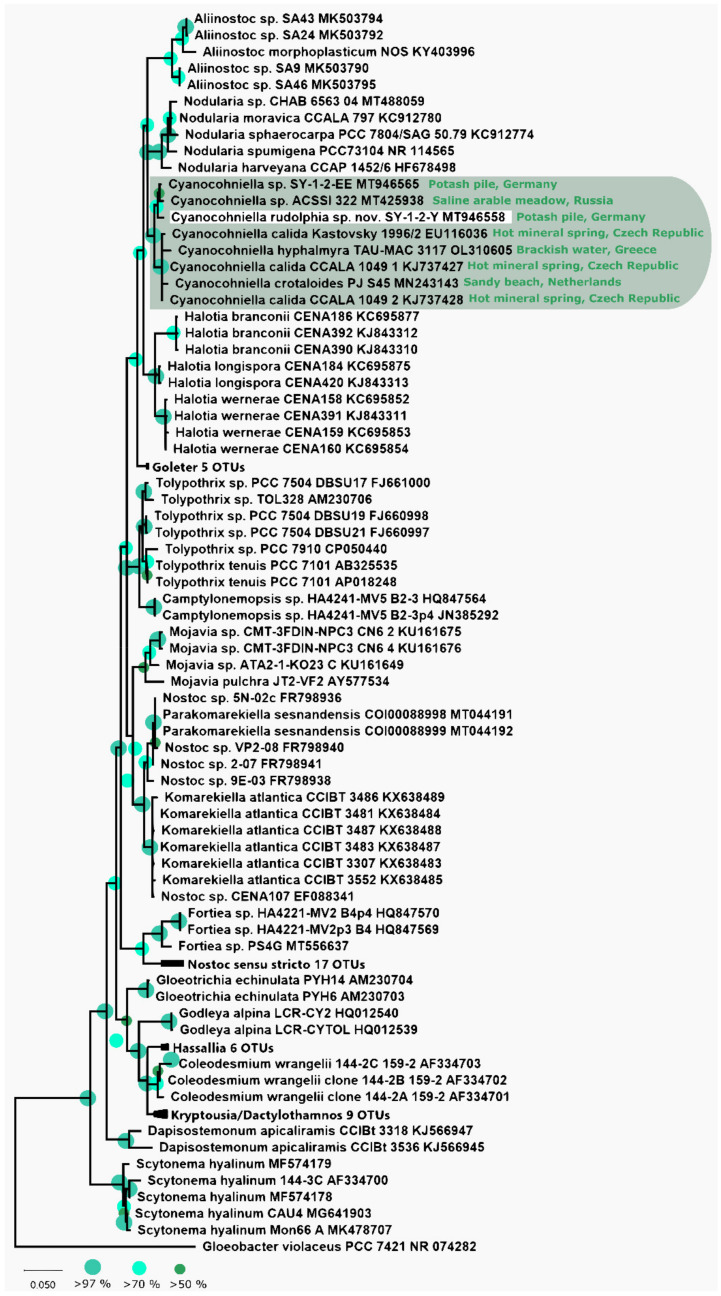
Phylogenetic maximum likelihood (ML) tree based on the 16S rRNA gene region. ML phylogenetic tree comprising 108 sequences (1.501 bp) of representatives of the order Nostocales rooted to *Gloeobacter violaceus*. Since the resulting Bayesian Interference (BI) and ML phylogenetic trees showed the same topology, a single tree with both BI and ML bootstrap values is shown. Supports at the nodes represent posterior probabilities and bootstrap values indicated as circles of different sizes and colors which refer to percent intervals.

**Figure 2 microorganisms-10-00968-f002:**
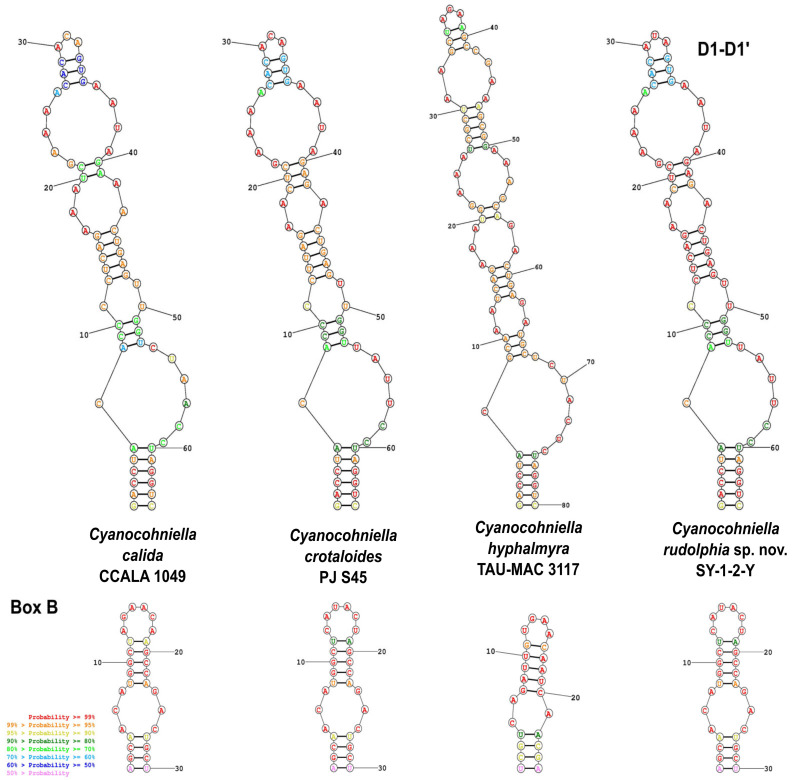
Predicted secondary structures. Shown are the predicted secondary structures of the D1-D1′ and Box B domains of the 16S-23S rRNA ITS gene region for all described *Cyanocohniella* species.

**Figure 3 microorganisms-10-00968-f003:**
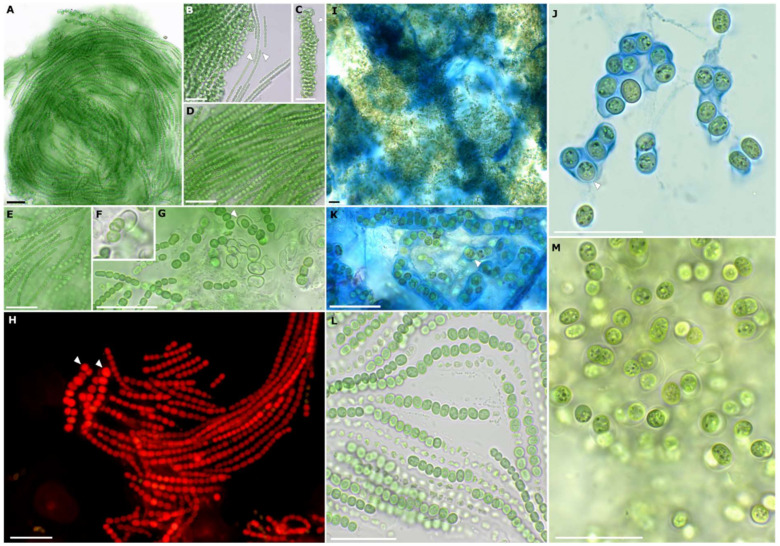
Morphological details of *Cyanocohniella rudolphia* sp. nov. (**A**) Overview of *Nodularia*-like adult stage with uni-seriate, straight and parallel arranged, mat-forming filaments. (**B**) *Pseudanabaena*/*Leptolyngbya*-like motile hormogonia with apical cells (arrows) and granulated cell content. (**C**) *Nostoc*-like stage formed by densely coiled uni-seriate filaments composed of smaller (compared to the *Nodularia*-like stage) cells that are embedded in a wide, hyline, limited and non-lamellate sheath (arrow). (**D**,**E**) details of *Nodularia*-like stage with hormogonium in (**E**) showing apical cells at both poles of the filament. (**F**,**G**) akinetes surrounded by firm and limited sheath (arrow) and filament development out of akinetes. (**H**) autofluorescence of straight *Nodularia*-like filaments and akinetes (arrows) with homogeneously fluorescing cells that indicate a coiled thylakoid distribution in the cells. (**I**–**K**) ACN staining visualizes the massive sheath formed by adult colonies in (**I**) and around akinetes with thick cell walls in (**J**,**K**) (arrows). (**L**) serial akinete formation in filaments surrounded by thick, limited and non-lamellate sheaths. (**M**) adult akinetes of older colonies that already disintegrated from the filaments. Cells are granulated and sometimes several cells are enclosed in a common thick, hyaline, non-lamellate and limited sheath. Scale bars indicate 25 µm.

**Figure 4 microorganisms-10-00968-f004:**
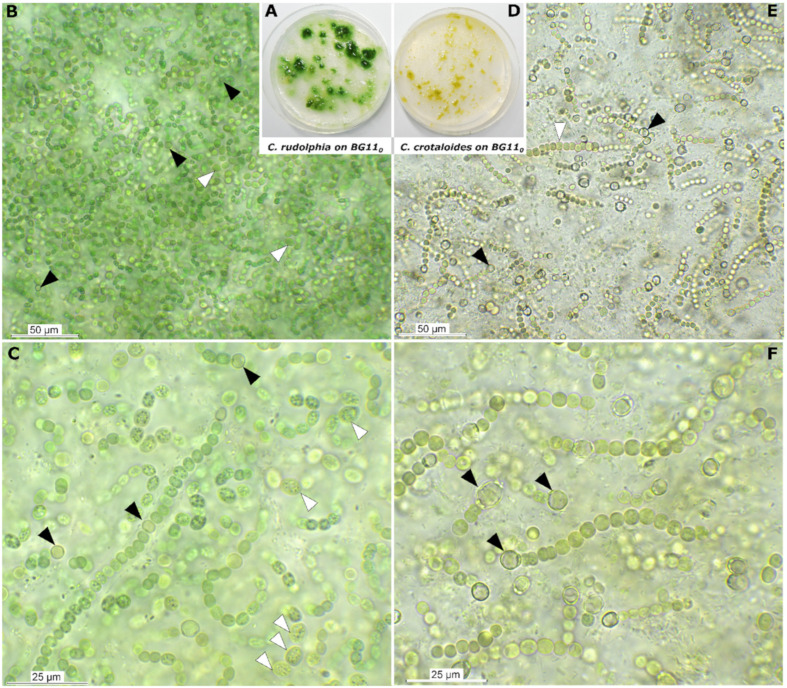
Comparison of *Cyanocohniella rudolphia* sp. nov. and *Cyanocohniella crotaloides* on nitrogen-free BG11_0_ medium after 6 weeks of growth. (**A**) Overview photograph of *C. rudolphia* on BG11_0_ with blue-green slimy colonies versus (**D**) yellowish faded colonies of *C. crotaloides*. (**B**,**C**) micrographs of *C. rudolphia* with slightly coiled, blue-green filaments, granulated akinetes (white arrows) and intercalar heterocytes (black arrows). (**E**,**F**) micrographs of *C. crotaloides* with non-granulated akinetes (white arrow) and short filaments with mostly polar heterocytes (black arrow) that show translucent cell content.

**Figure 5 microorganisms-10-00968-f005:**
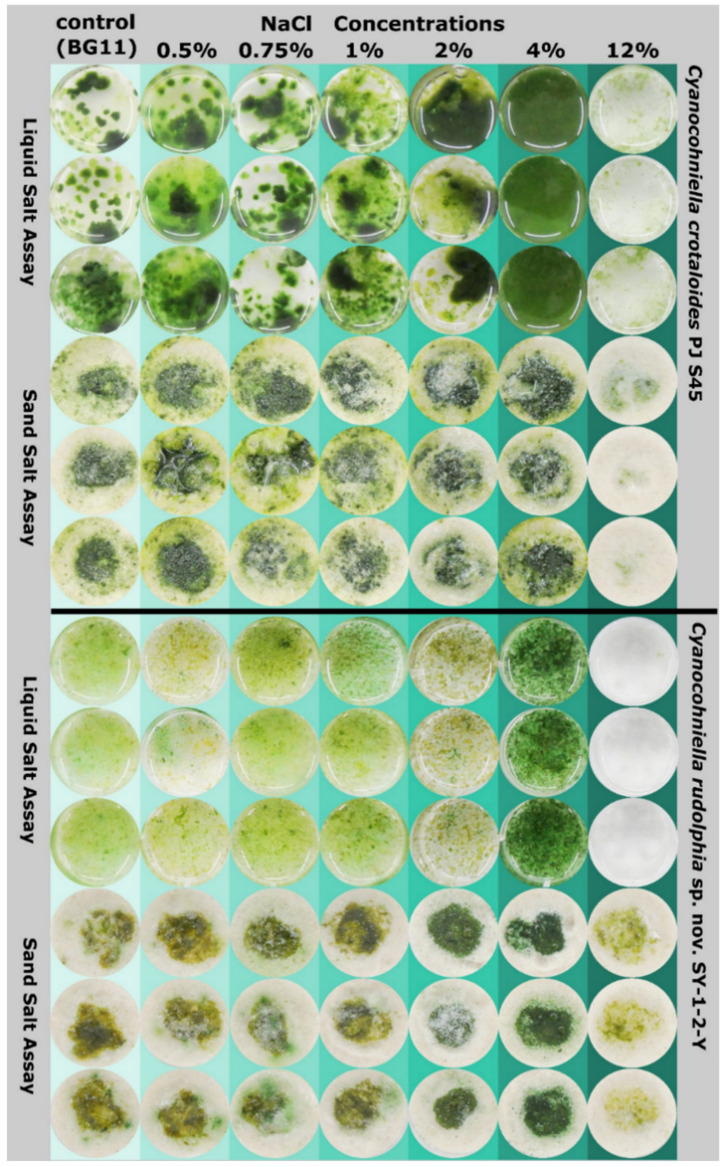
Photographs of salt assays after eight weeks of growth. Shown are photographs of wells containing liquid BG11 with various salt concentrations given in percent and biomass of *Cyanocohniella crotaloides* and *C. rudolphia*. The top three rows of each species show growth patterns in liquid medium (liquid salt assay) while the three bottom rows show growth patterns on the surface of sand (sand salt assay).

**Figure 6 microorganisms-10-00968-f006:**
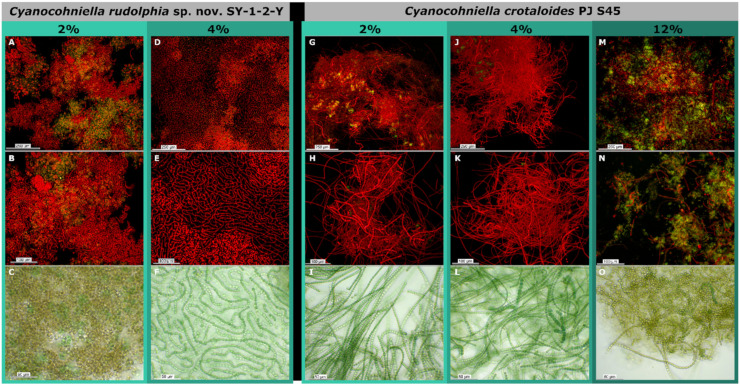
Micrographs of liquid salt assays after eight weeks of growth obtained by light microscopy and autofluorescence. (**A**–**C**) microscopy of *Cyanocohniella rudolphia* cultivated in 2% NaCl showing short filaments, unicellular cells that are detached from the filaments, akinete accumulation and fading photosynthetic pigments. (**D**–**F**) *C. rudolphia* cultivated in 4% NaCl with long, vividly colored, intact filaments, without akinetes. (**G**–**I**) *C. crotaloides* grown in 2% NaCl showing long, intact filaments with only a few decolorized cells and akinetes. (**J**–**L**) *C. crotaloides* grown in 4% NaCl with dense and long, intact and vividly colored filaments without akinetes. (**M**–**O**) *C. crotaloides* grown in 12% NaCl with fragmented filaments of pale color, degraded photosynthetic pigments and akinete accumulation.

**Figure 7 microorganisms-10-00968-f007:**
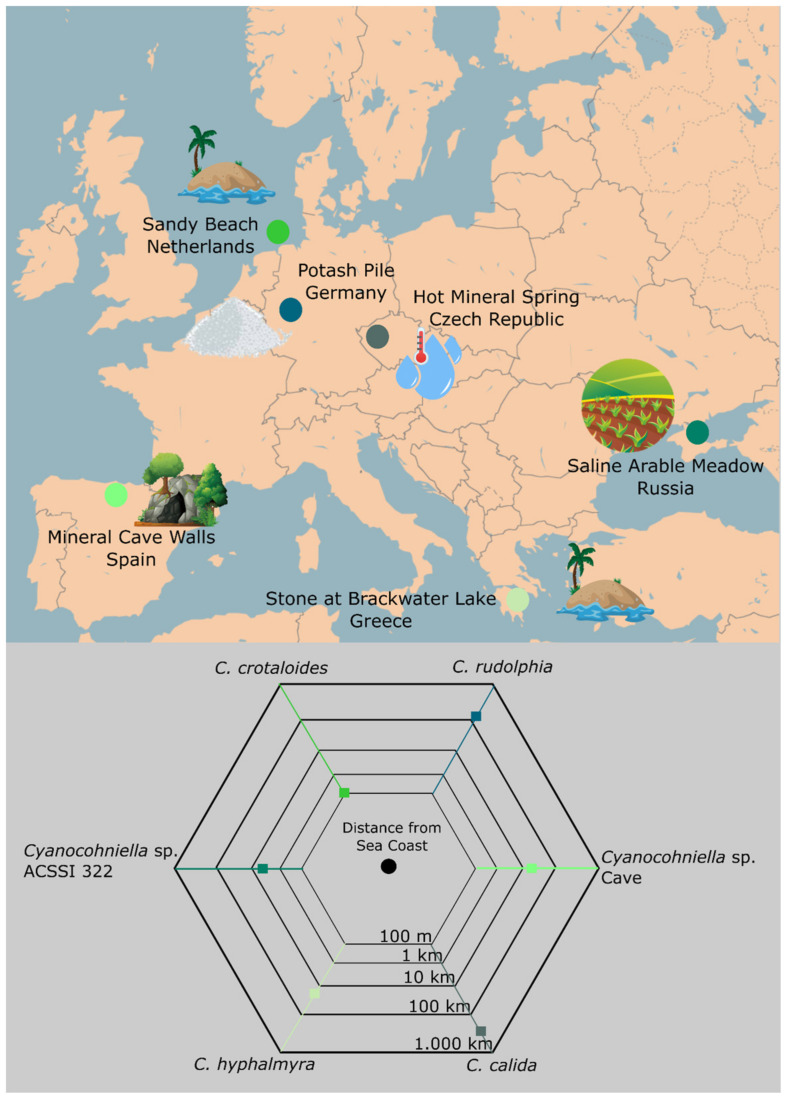
Distribution of *Cyanocohniella* strains. The map shows the countries and environments of isolated *Cyanocohniella* strains in addition to the distance to the coast.

**Table 1 microorganisms-10-00968-t001:** Mean photosynthetic quantum yield of PSII (*ΦPSII* = *F**′**_m_* − *F_t_*)/*F**′**_m_*) of chlorophyll *a* fluorescence of *Cyanocohniella crotaloides* and *Cyanocohniella rudolphia* (*n* = 3) cultivated as sand salt assay with the addition of liquid BG11 medium under varying salt conditions (0%, 0.5%, 0.75%, 1%, 2%, 4% and 12% NaCl).

NaCl Concentration (%)	Mean *ΦPSII* *C. crotaloides*	SD	Mean *ΦPSII* *C. rudolphia*	SD
Control	0.206	0.033	0.173	0.007
0.5	0.258	0.007	0.167	0.022
0.75	0.263	0.013	0.233	0.022
1	0.229	0.036	0.171	0.012
2	0.276	0.004	0.193	0.090
4	0.292	0.008	0.473	0.174
12	0.014	0.020	0.023	0.004

## Data Availability

The generated DNA sequences have been submitted to NCBI.

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
