# Peer review of "Salty Twins: Salt-Tolerance of Terrestrial Cyanocohniella Strains (Cyanobacteria) and Description of C. rudolphia sp. nov. Point towards a Marine Origin of the Genus and Terrestrial Long Distance Dispersal Patterns"

_microorganisms, 2022, doi:10.3390/microorganisms10050968_

Round 1

Reviewer 1 Report

This study presented the morphological and salt-tolerance physiological data for two terrestrial cyanobacterial strains (Cyanocohniella rudolphia and Cyanocohniella crotaloides), it also provided the description about C. rudolphia sp. nov.  The author demonstrated that the optimum salt condition for the growth was at 4% (500 mM) NaCl based mainly on the morphological observation and Chlorophyll-associated autofluorescence of the cell. Based on the phenomenon of colonization of salty habitats, the authors proposed the marine origin and expands colonization to salty terrestrial habitats.

Generally, the datum description is clear and the reviewer particularly likes the morphological pictures. The reviewer has no major concern, the following minor points should be addressed in the revision.

  1. The authors propose the marine origin of the studied Cyanocohniella rudolphia sp. nov. SY-1-2-Y. This is very interesting. The reviewer suggests that the authors also check the isolation origins of other relevant Cyanocohniellas strains that showed the high 16S rRNA identity to the strain SY-1-2-Y. Especially the Cyanocohniellas strains shown in phylogenetic tree of Figure 1. This would help us to understand expansion scenario of migration of marine strains towards the salty terrestrial habitat.
  2. Line 179-186: This section is about ACN staining, the corresponding reference is missing.
  3. Figure 3H: This is the autofluorescence subpicture of Cyanocohniella rudolphia, the corresponding brightfield subpicture is missing and should be presented here to help observe the morphology of akinete indicated by arrow in Figure 3H.
  4. Line 325-327: “Cell division takes place in one plane during all stages. Thylakoid position is coiled as the cell content is homogeneously distributed across the mostly non-granulated blue-green cells” which figure data support such conclusion? Did the authors conduct TEM work here? TEM picture should be provided.

Author Response

Response to Review for

‘Salty Twins: Salt-tolerance of terrestrial Cyanocohniella strains (Cyanobacteria) and description of C. rudolphia sp. nov. point towards a marine origin of the genus and terrestrial long distance dispersal patterns’ (microorganisms-1684376)

Reviewer I:

This study presented the morphological and salt-tolerance physiological data for two terrestrial cyanobacterial strains (Cyanocohniella rudolphia and Cyanocohniella crotaloides), it also provided the description about C. rudolphia sp. nov.  The author demonstrated that the optimum salt condition for the growth was at 4% (500 mM) NaCl based mainly on the morphological observation and Chlorophyll-associated autofluorescence of the cell. Based on the phenomenon of colonization of salty habitats, the authors proposed the marine origin and expands colonization to salty terrestrial habitats.

Generally, the datum description is clear and the reviewer particularly likes the morphological pictures. The reviewer has no major concern, the following minor points should be addressed in the revision.

The authors would like to thank the reviewer for taking his/her time to critically evaluate our manuscript. We are pleased to see the scientific contribution of this manuscript was ascertained and addressed all issues in the new version of the manuscript. The authors hope that the new version of the manuscript will soon be accepted by the reviewers and is suitable for a full publication in mdpi Microorganisms.

The authors propose the marine origin of the studied Cyanocohniella rudolphia sp. nov. SY-1-2-Y. This is very interesting. The reviewer suggests that the authors also check the isolation origins of other relevant Cyanocohniellas strains that showed the high 16S rRNA identity to the strain SY-1-2-Y. Especially the Cyanocohniellas strains shown in phylogenetic tree of Figure 1. This would help us to understand expansion scenario of migration of marine strains towards the salty terrestrial habitat.

This was now corrected and the full information about the ecological origin of all 16S rRNA Cyanocohniella sequences is given in the phylogenetic tree (fig. 1).

Line 179-186: This section is about ACN staining, the corresponding reference is missing.

ACN staining was originally designed for botanical thin sections of vascular plants, not for microorganisms. However, we recently applied this staining already in several other articles for cyanobacteria because we found that the chemical combination and staining effects are well suitable for the detection in differences of e.g. the EPS layer. We explicitly explained the percentual composition of the exact mixture thus we don’t think it is appropriate to give a reference for this widely used dye in botanical science.

Figure 3H: This is the autofluorescence subpicture of Cyanocohniella rudolphia, the corresponding brightfield subpicture is missing and should be presented here to help observe the morphology of akinete indicated by arrow in Figure 3H.

There was no corresponding brightfield image of this specific frame taken. The akinetes are depicted in brightfield in several other images such as fig. 3J, G, K, M and again in fig. 4B, C.

Line 325-327: “Cell division takes place in one plane during all stages. Thylakoid position is coiled as the cell content is homogeneously distributed across the mostly non-granulated blue-green cells” which figure data support such conclusion? Did the authors conduct TEM work here? TEM picture should be provided.

Autofluorescence is equally distributed across the cell content which points towards a coiled thylakoid arrangement. If they were parietal the main fluorescence would be arranged close to the cell walls, what is not the case here. TEM pictures are for sure the ultimate methods to evaluate the thylakoid positioning in cyanobacteria but they are not available for any other Cyanocohniella species and hence a comparison e.g. with the type strain is not possible. For these reasons the authors did not include TEM images but instead gave information about the thylakoid arrangement by means of autofluorescence, a technique to which more people have access then to TEM. In addition, thylakoid arrangements have now repeatedly been shown to be not suitable as a distinct characteristic for taxonomic purposes, since their arrangement is not consistently expressed across genera or even species within one genus.

Reviewer 2 Report

Manuscript title“ Salty Twins: Salt-tolerance of Terrestrial Cyanocohniella 2

Strains (Cyanobacteria) and Description of C. rudolphia sp. 3 nov. Point towards a Marine Origin of the Genus and Terres- 4 trial Long Distance Dispersal Patterns” is a well-written manuscript with extensive work. Paper could be very interesting for the reader of sustainability. I have few points to be clear in the paper, and those should be clarified before the paper would be accepted.

Abstract

  1. Abstract not consist any Physiological feature of salt-tolerant cyanobacteria whereas extensively worked on it in the manuscript.
  2. Abstract should have crisp information about the aim materials method result, and conclusion, which I don‘t find in the present abstract form.
  3. It always crucial to consider ecological importance also of the proposed manuscript which I find a missing link

Introduction

  1. In The Introduction there is no any Recent studies incorporated to give clarity about the other factors and salt in corelation with cyanobacteria diversity and morphological variations.

  1. Is the Salt only one factor affects the diversification with respect to Ecological Niche.
  2. There should be more updated information in relation with the findings of the research.
  3. Is there any information about the sampling site temperature, salt and community relationship? Because temperature cannot be ignored.

Material and Method

  1. Sampling procedure should be very well written as this part is the base point of the project, where other elemental concentration also cannot be ignored.
  2. DNA isolation and quantification have done just after the sampling or isolated from old stored sample?
  3. Line 49-59 kindly add more specific molecular or biochemical features of cyanobacteria, due to which these microorganisms have higher acclimatization capabilities with recent and advance information. Eg: unique expression of genes, Metabolisms or unique protein synthesis involve in stress tolerance capability
  4. Line 70-Why only sucrose ( other carbon molecules?) accumulation came in the picture suddenly. Fist need to explain the role of the sucrose then specify the findings,

Result

  1. Phylogenetic analysis is very well written just one table need to add for morphological changes and growth pattern with respect to Salt Concentration.
  2. Figure 3 not clear which model of microscope and camara at what specification cells are measured??
  3. validate your points about changes in cell physiology with high salt stress
  4. There is no information available in the paper about salt tolerance capabilities with respect to sucrose accumulation.
  5. Add few more point for dispersal of the model organism of the study?

Conclusion

  1. The corelation between different factors available in Stress site and microbial community should be clearly written

Discussion

  1. Discussion part should have more advance and recent references to validate and make more interesting you findings.
  2. Previous studies on dominating microbes in saline soil and in relation to your findings also ecological application of the findings.
  3. What are the other factors based on previous studies, can be the part of you finding?
  4. More Recent references should be used to discuss your findings? With the future possibilities.

Author Response

Response to Review for

‘Salty Twins: Salt-tolerance of terrestrial Cyanocohniella strains (Cyanobacteria) and description of C. rudolphia sp. nov. point towards a marine origin of the genus and terrestrial long distance dispersal patterns’ (microorganisms-1684376)

Reviewer II

Manuscript title“ Salty Twins: Salt-tolerance of Terrestrial Cyanocohniella 2 Strains (Cyanobacteria) and Description of C. rudolphia sp. nov. Point towards a Marine Origin of the Genus and Terrestrial Long Distance Dispersal Patterns” is a well-written manuscript with extensive work. Paper could be very interesting for the reader of sustainability. I have few points to be clear in the paper, and those should be clarified before the paper would be accepted.

The authors would like to thank the reviewer for taking his/her time to critically evaluate our manuscript. We are pleased to see the scientific contribution of this manuscript was ascertained and addressed all issues in the new version of the manuscript. The authors hope that the new version of the manuscript will soon be accepted by the reviewers and is suitable for a full publication in mdpi Microorganisms.

Abstract

Abstract not consist any Physiological feature of salt-tolerant cyanobacteria whereas extensively worked on it in the manuscript.

Abstract should have crisp information about the aim materials method result, and conclusion, which I don‘t find in the present abstract form.

It always crucial to consider ecological importance also of the proposed manuscript which I find a missing link

The abstract was slightly re-arranged and includes information about the ecological and biotechnological importance of our findings. The amount of information that can be given here is strictly limited by the publishing journal and thus cannot be worked out extensively.

Introduction

In The Introduction there is no any Recent studies incorporated to give clarity about the other factors and salt in corelation with cyanobacteria diversity and morphological variations. Is the Salt only one factor affects the diversification with respect to Ecological Niche.

Various factors influence a cyanobacterial community. In the case of Cyanocohniella it was previously assumed that this might be a thermophilic lineage which has already proven not to be true. This fact has already been discussed previously (Jung et al., 2020, Panaus and Gkelis 2021) and is also mentioned in the manuscript. This again demonstrates that it is very hard to find a strict niche concept for most cyanobacteria, which is discussed in various chapters of the manuscript.

There should be more updated information in relation with the findings of the research.

The findings of our study are presented in the results part and are discussed in depth in the discussion part. As the literature body on e.g. morphological changes under varying salt concentrations is limited, not all information can directly be given in the introduction because this would result in a repetition in the discussion part. For these reasons more information is now given in the discussion part.

Is there any information about the sampling site temperature, salt and community relationship? Because temperature cannot be ignored.

Research on this study site has taken place during the last couple of years where an extensive research body was created. This research gives deep insights into the community structure of the site addressing all of these aspects. This literature is cited throughout the manuscript and is already introduced to the reader in the introduction (see Sommer et al., 2022; Pushkareva et al., 2021; Sommer et al., 2020a; Sommer et al., 2020b)

Material and Method

Sampling procedure should be very well written as this part is the base point of the project, where other elemental concentration also cannot be ignored.

Sampling procedure was not part of this study and is described for C. rudolphia in Sommer et al., 2020a and for C. crotaloides in Jung et al., 2020 which is already stated in the methods section. Also the isolation process until unialgal cultures were achieved is well given in the two corresponding articles. In these two articles more information is also presented about the ecology of the two sites.

DNA isolation and quantification have done just after the sampling or isolated from old stored sample?

DNA isolation was carried out from fresh culture material which is now stated in the manuscript.

Line 49-59 kindly add more specific molecular or biochemical features of cyanobacteria, due to which these microorganisms have higher acclimatization capabilities with recent and advance information. Eg: unique expression of genes, Metabolisms or unique protein synthesis involve in stress tolerance capability

This section is in the introduction. It was slightly modified but we do not want to go deeper into metabolism or gene expression here as this is off the scope of the article. The new section now reads: “Besides the well-known cyanobacterial harmful algal blooms (cHABs) that can be monitored from space due to their large distribution (Fang et al. 2019), cyanobacteria are also common pioneer organisms in desert soils (Belnap 2003), colonizers of Ant-arctica (Pushkareva et al. 2018), cave inhabitants (Behrendt et al. 2020), symbiotic partners of lichens (Jung et al. 2021) and are even presumed to be ancient life forms on Mars (Joseph et al., 2020; (Billi et al. 2019). This broad spectrum of habitat, geographic distance and abiotic conditions under which cyanobacteria can thrive defining their ecological niche is reached by their high acclimatization capability. Rapid changes in gene regulation in cyanobacteria that affect the expression of such as the one encoding molecular chaperones, photosynthetic and oxidative stress-related genes followed by complex metabolic adjustments are part of their ecological success story (Rosic 2021) (Singh 2018). Besides drastic temperature amplitudes (Lakatos und Strieth 2017), long periods of desiccation, strong radiation and pH extremes (Chen et al. 2021) they are also able to flourish in saline environments  (Oren 2015).

Line 70-Why only sucrose ( other carbon molecules?) accumulation came in the picture suddenly. Fist need to explain the role of the sucrose then specify the findings,

This section is in the introduction. It was now changed according to the suggestions of the reviewer and now reads: “Their basal salt acclimation strategy includes two principal responses, the active export of metabolically harmful ions (mainly sodium and chloride) and the accumulation of compatible solutes via biosynthesis or uptake from the environment (Hagemann, 2011). Primarily, the four major compatible solutes sucrose, trehalose (sugars), glucosylglycerol (heterosides, Hagemann and Pade, 2015), and glycine betaine (amino acid derivates) are synthesized and/or accumulated by cyanobacteria (Hagemann, 2011). These low-molecular mass substances are readily soluble in water, can be accumulated in high concentrations and do not interfere with the cell’s metabolism but compensate for changes in the osmotic potential (Kirsch et al., 2019). One example is the strain Desmonostoc salinum CCM-UFV059 that showed a salt tolerance when compared to the model Nostoc sp. PCC 7120 by accumulating sucrose as the main compatible solute under saline conditions as...”

Result

Phylogenetic analysis is very well written just one table need to add for morphological changes and growth pattern with respect to Salt Concentration.

We only detected differences in vitality of the two strains under varying salt concentrations and only minor observations in terms of morphological changes. This is now further extended in the discussion.

Figure 3 not clear which model of microscope and camara at what specification cells are measured??

Detailed information on the type of microscope, the camera etc. is given in the methods section: “Morphological characteristics were evaluated with a Panthera KU trinocular light microscope (Motic, Barcelona, Spain) coupled with a MicroLive Multi Format camera and the software MicroLive (v4.0). About 200 cells were measured, and a mean size range is given, reflecting the variability of irregular cell sizes. In addition to bright-field images, samples were analyzed under autofluorescence and ACN staining. The latter is a 20:1:1 mix of Astra Blue, chrysoidin, and Neufuchsin (0.1 g Astra Blue in 79.5 ml H2O and 2.5 ml acetic acid, 0.1 g chrysoidin in 100 ml H2O, and 0.1 g Neufuchsin in 100 ml H2O) that allows a differentiation of structures according to color due to the binding characteristics of the substances.” All scale bars indicate 25 µm as can be taken from the caption of the images.

validate your points about changes in cell physiology with high salt stress

This is now discussed in the discussion part of the manuscript in more depth.

There is no information available in the paper about salt tolerance capabilities with respect to sucrose accumulation.

Sucrose is only one of the produced compatible solutes as outlined in the introduction. For the genus Cyanocohniella it is not known which compatible solute/substance is produced and/or involved in salt stress. For these reasons no information can be given at this stage. In addition such information belongs to the introduction or discussion of a manuscript, not into the results section if no experiment was conducted investigating sucrose.

Add few more point for dispersal of the model organism of the study?

All available information on the genus Cyanocohniella is given within the manuscript. All available 16S rRNA gene sequences of Cyanocohniella and related but yet undescribed strains are addressed in the manuscript, including the origin of these strains. So far members of this genus have only been found where indicated in the article.

Conclusion

The corelation between different factors available in Stress site and microbial community should be clearly written

 We already have a follow-up study running where the genome of the strains will be sequenced. For these reasons we would like to keep the conclusion short and focused on the information that can be gained from the genome with respect to the distribution patterns of the genus Cyanocohniella.

Discussion

Discussion part should have more advance and recent references to validate and make more interesting you findings.

We now updated especially the part discussing morphological changes upon increasing salt concentrations. The new part now reads: “Also fluorescence microscopy and light microscopy revealed that the populations in the liquid salt assay were most intact with little akinete formation (Fig. 6) as a sign of unfavorable culture conditions. During comparable studies Fremyella diplosiphon and also Anabaena cylindrica, for example, showed morphological abnormalities induced by salt that are most likely caused by salt-induced osmotic stress (Sing et al., 2013; Bhadauriya et al., 2007). For, Synechococcus sp. PCC 7942 for example, it was detected that salt concentrations of 500 mM triggered cell elongation and resulted in bending of cells (Verma et al., 2018). These results are in agreement with the observations of Meur (1988) and Piuri et al. (2005) in E. coli and Lactobacillus casei, respectively, where salt stress triggers the cell size elongation.”

Previous studies on dominating microbes in saline soil and in relation to your findings also ecological application of the findings. What are the other factors based on previous studies, can be the part of you finding? More Recent references should be used to discuss your findings? With the future possibilities.

We now added an additional paragraph to the discussion that addresses biotechnological benefits as well as ecological implications. The new paragraph reads: Ecological Importance and Biotechnology

To increase sustainability, large-scale cultivation of cyanobacteria for biotechnological production of biofuels or bulk chemicals it should be avoided to use large amounts of limited freshwater resources. Instead mass cultivation of cyanobacteria should preferably be carried out in brackish or seawater (Chisti, 2013). The influence of salinity on cyanobacterial productivity and specific substances of high-value is scarcely investigated but strains that can potentially grow in sea water offer new opportunities for biotechnology. Here, some compatible solutes such as the heteroside glucosylglycerol can be used as stabilizing agent for enzymes or antibodies, and hence enables their long-term storage in freezers (Luley-Goedl et al., 2010; Tan et al., 2015) or is used in skin revitalizing creams (e.g. bitop AG, Germany, and designated as Glycoin®). With respect to productivity both Cyanocohniella strains investigated stand out as they reached best vitality under 4 % NaCl while the model strains widely used in biotechnology Synechocystis sp. PCC 6803 and Synechococcus sp. PCC 7002 showed decreased production rates of glucosylglycerol (Kirsch et al., 2019).

Besides biotechnology especially such salt tolerant terrestrial strains can play significant roles in remediation of (salt affected) soils where they can restore the function of the overall microbiome, prevent erosion or desertification (Li et al., 2019). For these purposes often the non-nitrogen fixing, filamentous cyanobacterium Microcoleus spp. is used (Giraldo-Silva et al., 2019) that showed great abilities in concatenating soil particles as an pioneer microbe. Recently, similar approaches have been undertaken with eukaryotic green algae that were used as inoculum in order to colonize and stabilize potash piles (Sommer et al., 2022). However, the two Cyanocohniella strains are also suitable for such purposes and additionally have the benefit that they can fix atmospheric nitrogen what enriches soil fertility and thus can speed up remediation processes.

Round 2

Reviewer 2 Report

The author has modified the manuscript as per my suggestion. I suggest accepting the manuscript in its present form.